# Concentration Dependence of Anti- and Pro-Oxidant Activity of Polyphenols as Evaluated with a Light-Emitting Fe^2+^-Egta-H_2_O_2_ System

**DOI:** 10.3390/molecules27113453

**Published:** 2022-05-27

**Authors:** Michal Nowak, Wieslaw Tryniszewski, Agata Sarniak, Anna Wlodarczyk, Piotr J. Nowak, Dariusz Nowak

**Affiliations:** 1Radiation Protection, University Hospital No. 2, Medical University of Lodz, Zeromskiego 113, 90-549 Lodz, Poland; m.nowak@skwam.lodz.pl; 2Department of Radiological and Isotopic Diagnostics and Therapy, Medical University of Lodz, Zeromskiego 113, 90-549 Lodz, Poland; wieslaw.tryniszewski@umed.lodz.pl; 3Department of Clinical Physiology, Medical University of Lodz, Mazowiecka 6/8, 92-215 Lodz, Poland; agata.sarniak@umed.lodz.pl; 4Department of Sleep Medicine and Metabolic Disorders, Medical University of Lodz, Mazowiecka 6/8, 92-215 Lodz, Poland; anna.wlodarczyk@umed.lodz.pl; 5Department of Nephrology, Hypertension, and Kidney Transplantation, Medical University of Lodz, Pomorska 251, 92-213 Lodz, Poland; piotr.nowak@umed.lodz.pl

**Keywords:** polyphenols, plant phenolic acids, chemiluminescence, Fenton system, antioxidant activity, pro-oxidant activity, catechol, methoxyphenol

## Abstract

Hydroxyl radical (^•^OH) scavenging and the regeneration of Fe^2+^ may inhibit or enhance peroxidative damage induced by a Fenton system, respectively. Plant polyphenols reveal the afore-mentioned activities, and their cumulative net effect may determine anti- or pro-oxidant actions. We investigated the influence of 17 phenolics on ultra-weak photon emission (UPE) from a modified Fenton system (92.6 µmol/L Fe^2+^, 185.2 µmol/L EGTA (ethylene glycol-bis(β-aminoethyl-ether)-N,N,N′,N,-tetraacetic acid) and 2.6 mmol/L H_2_O_2_ pH = 7.4). A total of 8 compounds inhibited (antioxidant effect), and 5 enhanced (pro-oxidant effect) UPE at all studied concentrations (5 to 50 µmol/L). A total of 4 compounds altered their activity from pro- to antioxidant (or vice versa) along with increasing concentrations. A total of 3 the most active of those (ferulic acid, chlorogenic acid and cyanidin 3-*O*-glucoside; mean UPE enhancement by 63%, 5% and 445% at 5 µmol/L; mean UPE inhibition by 28%, 94% and 24% at 50 µmol/L, respectively) contained catechol or methoxyphenol structures that are associated with effective ^•^OH scavenging and Fe^2+^ regeneration. Most likely, these structures can determine the bidirectional, concentration-dependent activity of some phenolics under stable in vitro conditions. This is because the concentrations of the studied compounds are close to those occurring in human fluids, and this phenomenon should be considered in the case of dietary supplementation with isolated phenolics.

## 1. Introduction

Numerous phytochemicals, including phenolics, are recognized as potent antioxidants that may contribute to the health benefits of diets rich in fruits and vegetables [1,2]. The antioxidant effect of ingested polyphenols is multidirectional; they can stimulate the biosynthesis of antioxidant enzymes [3], suppress processes leading to the production of reactive oxygen species (ROS) in vivo [3,4] or directly inhibit or decompose ROS in chemical reactions [5]. Direct antioxidant activity may involve reactions with ROS and reactive nitrogen species (e.g., hydroxyl radicals, superoxide radicals, nitric oxide or peroxynitrite) to form unreactive products and the chelation of transition metals ions (Fe^2+^, Fe^3+^, Cu^2+^) to prevent their reactions with H_2_O_2_, leading to the generation of hydroxyl radicals (^•^OH) [5,6,7]. The presence of various chemical groups, as well as their number and localization in the backbone structure (e.g., catechol structure, hydroxy group (-OH) and carboxyl group (-COOH)), may be responsible for these properties, but the occurrence of conjugated double bonds and the nature of aliphatic substituents on the benzene ring may also affect these properties [8,9,10].

On the other hand, plant polyphenols can also function as pro-oxidants, and the chemical groups contributing to the antioxidant activity of certain polyphenols may be predisposed to pro-oxidant effects in the case of other compounds, even if studied under the same conditions [9]. For instance, in the group of five phenolics tested at a concentration of 10 µmol/L, which inhibited the oxidation of deoxyribose by the Fe^2+^- EDTA (ethylenediaminetetraacetic acid) -H_2_O_2_ system (antioxidant effect), this activity, expressed as % inhibition, was positively correlated with the number of -OH groups; however, for the other eight phenolics with pro-oxidant activity (% enhancement), the same association was found [9].

Concentration changes in a given polyphenol under fixed conditions of a redox system are another issue. One may expect that raising the concentration of a compound with antioxidant activity results in a stronger antioxidant effect. This is based on the assumption of an almost steady relationship between the activities of a particular polyphenol’s chemical groups and compound concentration. Thus, if we use a compound with antioxidant activity at higher concentrations, the ROS suppression is stronger. Otherwise, pro-oxidant activity at higher concentrations exerts a stronger pro-oxidant effect. However, activities or even properties of chemical groups of a given polyphenol may alter, depending on the compound concentration. For instance, the degree of dissociation of a -COOH group in a phenolic ring or in a hydrocarbon substituent decreases with increasing concentrations of a given polyphenol. The deprotonation of this group and the negative charge of polyphenol molecule may modify the reactivity of other groups. Therefore, it cannot be excluded that a compound presenting antioxidant activity at a given concentration may lose these properties or even become pro-oxidant at another one. This assumption agrees with the observation that Trolox, within the range of concentrations from 2.5 µmol/L to 15 µmol/L, reveals an antioxidant effect in HeLa cells culture, whereas at higher concentrations, from 40 µmol/L to 160 µmol/L, it is pro-oxidant [11]. Trolox is a water-soluble analog of vitamin E, having a carboxyl group at the heterocyclic ring of the chroman head instead of a long alkyl chain [12]. Numerous polyphenols (e.g., cyanidin, malvidin, rutin and quercetin) tested in a wide range of concentrations, from 50 µmol/L to 4000 µmol/L, with the HPLC method (measurement of malondialdehyde generated during the oxidation of linoleic acid emulsion exposed to Cu^2+^ and ambient air) have revealed dual distinct pro-oxidant and antioxidant activity, consisting of the enhancement or suppression of malondialdehyde formation, respectively [13]. In another report, the aqueous extract of polyphenols from rooibos (*Aspalathus linearis*) at a concentration of 0.1 mg/mL enhanced Fe^3+^-EDTA-H_2_O_2_ -induced deoxyribose degradation, whereas at a 15-times higher concentration, significant inhibition was noted, indicating the net scavenging of ^•^OH radicals [14]. Aspalathin and isovitexin were the major polyphenols in this extract [14], which suggests concentration dependency of their anti- and pro-oxidant potentials. However, the tested concentrations of pure polyphenols and polyphenol extracts were many times higher than those occurring in human plasma [15], which is a serious limitation of the aforementioned studies. Recently, we developed a Fenton’s-reaction-based system, emitting ultra-weak chemiluminescence. It was composed of Fe^2+^, EGTA (ethylene glycol-bis (β-aminoethyl ether)-N,N,N′,N′,-tetraacetic acid) and H_2_O_2_. ^•^OH radicals, generated in the reaction of Fe^2+^ with H_2_O_2_, can attack ether bonds in EGTA molecules, leading to the formation of excited carbonyl groups and then to ultra-weak photon emission (UPE) [16]. Moreover, quenching or enhancing light emission from this system may serve as an indicator of anti- or pro-oxidant activities of a given compound, including plant polyphenols and antioxidant vitamins [16,17]. Therefore, in this study, we decided to evaluate the effects of 17 plant polyphenols and Trolox on the UPE of a Fe^2+^-EGTA-H_2_O_2_ system. The objectives of this study were: (A) to evaluate the anti- and pro-oxidant activities of several polyphenols within the concentration range of 5 µmol/L to 50 µmol/L, which may occur in human body fluids after the consumption of a meal rich in polyphenols; (B) to verify the hypothesis that antioxidant activity may convert into pro-oxidant activity (or vice versa), depending on the concentration of the tested compound; and (C) to find any relationships between compound structures and their effects on light emission from the Fe^2+^-EGTA-H_2_O_2_ system.

## 2. Results

The UPE of a 92.6 µmol/L Fe^2+^-185.2 µmol/L EGTA-2.6 mmol/L H_2_O_2_ system was 3057 ± 1057 (2766; 1057) RLU (n = 21). The light emissions from the incomplete control systems of Fe^2+^-H_2_O_2_ and Fe^2+^-EGTA were significantly lower (*p* < 0.05, n = 21) and reached 1118 ± 397 (1032; 366) RLU and 903 ± 416 (832; 178) RLU, respectively. These results are similar to those previously described [16,17], which also showed no significant differences between light emissions from EGTA-H_2_O_2_ and H_2_O_2_ alone in comparison to the medium alone. For all tested compounds, there were no differences between light emissions from the compound-Fe^2+^-H_2_O_2_ and Fe^2+^-H_2_O_2_ samples. There were also no differences between the mean light signals emitted from the compound-Fe^2+^-EGTA and Fe^2+^-EGTA samples and from the compound in the medium alone.

A total of 8 of the 18 tested compounds inhibited light emission from the Fe^2+^-EGTA-H_2_O_2_ system within the range of concentrations, from 5 µmol/L to 50 µmol/L. A total of 5 compounds enhanced, and the other 5 revealed bidirectional (enhancing or inhibiting) activity. Since the UPE of Fe^2+^-EGTA-H_2_O_2_ depends on ^•^OH-radical-induced conversion of EGTA into derivatives with excited carbonyl groups [16,17], one may conclude that the inhibition or enhancement of light emissions may be a measure of the antioxidant and pro-oxidant activity of the tested compound, respectively.

### 2.1. Polyphenols with Antioxidant Activity within the Concentration Range of 5 µmol/L to 50 µmol/L

Table 1 shows the antioxidant activity of eight phenolics evaluated with the Fe^2+^-EGTA-H_2_O_2_ test in descending order. 3,4-dihydroxyphenylacetic acid and orthocresol, already at concentrations of 5 µmol/L, almost completely inhibited the UPE of the Fe^2+^-EGTA-H_2_O_2_ system. Moderate suppression (mean inhibition ranged from 46% to 30% at concentration of 5 µmol/L) of the UPE was noted for homovanillic acid, vanillic acid and caffeic acid. The lowest antioxidant activity was found for 4-hydroxy phenyl acetic acid, 3-hydroxybenzoic acid and hippuric acid. In addition, the last two compounds, even at concentrations of 50 µmol/L, did not inhibit more than one third of light emission from the Fe^2+^-EGTA-H_2_O_2_ system. Of six compounds causing moderate or weak UPE suppression at concentrations of 5 µmol/L, only caffeic acid and 4-hydroxy phenyl acetic acid revealed a positive, almost linear relationship between compound concentration and antioxidant effects (Table 1).

### 2.2. Polyphenols with Pro-Oxidant Activity within the Concentration Range of 5 µmol/L to 50 µmol/L

A total of 5 of the 18 tested compounds (gallic acid, phloroglucinol, pelargonidin, ellagic acid and pelargonidin-3-*O*-rutinoside) revealed pro-oxidant activity (Table 2). At concentrations of 5 µmol/L, the highest mean % enhancement of light emission from the Fe^2+^-EGTA-H_2_O_2_ system was found for gallic acid (1689 ± 358%), and the lowest one (but still significant) was observed for pelargonidin-3-*O*-rutinoside (75 ± 7%). A distinct positive linear relationship between compound concentration and the % enhancement of UPE was observed only for phloroglucinol and ellagic acid (Table 2).

### 2.3. Polyphenols Which Altered Their Antioxidant Activity into Pro-Oxidant Activity (or Vice Versa) within the Concentration Range of 5 µmol/L to 50 µmol/L

Three phenolics, ferulic acid, chlorogenic acid and cyanidin 3-*O*-glucoside at concentrations of 5 µmol/L, enhanced the UPE of the Fe^2+^-EGTA-H_2_O_2_ system. Both ferulic acid and cyanidin 3-*O*-glucoside at concentrations of 25 µmol/L still enhanced UPE (but the effect was significantly lower), and at concentrations of 50 µmol/L, they revealed an antioxidant effect. A decrease in light emission from the modified Fenton system was noted (Table 3). However, for chlorogenic acid, “the break point” occurred earlier; concentrations of 25 µmol/L and 50 µmol/L were antioxidant. Trolox, which served as a positive control, behaved in the same manner. Resorcinol was antioxidant at a concentration of 5 µmol/L, whereas at a higher concentration, it revealed a pro-oxidant effect. It should be mentioned that, for ferulic acid and resorcinol, a linear relationship between compound concentration and effect on light emission was noted (Table 3).

## 3. Discussion

Plant polyphenols can act as antioxidants or pro-oxidants depending on the kind of studied compound and are used in Fenton systems in vitro [9,18]. For instance, myricetin inhibited deoxyribose degradation by the Fe^3+^-EDTA-H_2_O_2_-ascorbic acid system and enhanced this process by Fe^3+^-EDTA. On the other hand, tricetin (which differs from myricetin in the lack of -OH group on ring C at C_3_) also protected deoxyribose from oxidative damage induced by Fe^3+^-EDTA-H_2_O_2_-ascorbic acid, whereas it had a very weak pro-oxidant effect in the case of Fe^3+^-EDTA [18]. In another study, the effect of 13 phenolics on Fe^2+^-EDTA-H_2_O_2_-induced deoxyribose degradation was evaluated; 4 of them significantly protected deoxyribose (antioxidant effect), and 7 revealed significant pro-oxidant activity. In addition, it is interesting that phenolics having remarkably similar structure revealed the opposite activity. 3,4-dihydroxycinnamic acid has a longer aliphatic substitute (just one carbon atom) on the catechol ring than 3,4-dihydroxyphenylacetic acid. However, 3,4-dihydroxycinnamic acid inhibited deoxyribose degradation, whereas the latter one enhanced the oxidative damage to this monosaccharide [9]. Our results confirm these observations but additionally show that a given phenolic tested under fixed conditions of a modified Fenton redox system can function as pro- or antioxidant, depending on its concentration. It should be pointed out that studied phenolic concentrations ranging from 5 µmol/L to 50 µmol/L can occur in systemic circulation or in blood in the portal vein after the ingestion of meals rich in plant phenolics [15]. Moreover, this phenomenon seems not to be unique because 4 of the 17 tested phenolics behaved in such a manner. There are numerous redox systems in the human body (e.g., cysteine/cystine (Cys/CySS), reduced glutathione/oxidized glutathione (GSH/GSSG), reduced nicotinamide adenine dinucleotide phosphate/oxidized nicotinamide adenine dinucleotide phosphate (NADPH/NADP^+^), Fe^2+^/Fe^3+^ and Cu^+^/Cu^2+^) [19,20,21,22,23], and many plant phenolics can interact with them at concentrations that can change depending on the type of consumed food [15]. Some compounds may inhibit, and other phenolics can augment peroxidative processes in the human body. Moreover, it is suggested that antioxidants, including plant polyphenols especially ingested in supraphysiological doses, may act as pro-oxidants; can augment oxidative damage to a variety of biomolecules (DNA, proteins and lipids); and can switch on the intracellular signaling pathways leading to the stimulation of an inflammatory response [24]. These clearly show that the issue of dietary supplementation with plant phenolics, in order to suppress oxidative processes in the human body, is very complicated and still requires controlled clinical trials. On the other hand, diets rich in fruits and vegetables have beneficial effects on human health, and the antioxidant and anti-inflammatory effects of ingested polyphenols is one of mechanisms explaining these epidemiological data [24]. Therefore, so far, the optimal prophylactic action is to increase the ingestion of polyphenols as well as vitamins, minerals and dietary fibers with a natural food matrix rather than the supplementation of the diet with one or more purified polyphenols.

### 3.1. Plausible Mechanism by Which Polyphenols May Affect the UPE of the Fe^2+^-EGTA-H_2_O_2_ System

The proposed mechanism of light emission by the Fe^2+^-EGTA-H_2_O_2_ system was described in detail in our previous study [16]. A summary of the chemical reaction leading to ^•^OH radical generation in our Fenton system is given below:Fe^2+^-EGTA + H_2_O_2_ → Fe^3+^-EGTA + OH^−^ + ^•^OH

^•^OH radicals generated in the Fenton reaction can attack one of the ether bonds in the backbone structure of EGTA, leading to its cleavage and to the generation of peroxyl radicals. They can react with each other, with subsequent formations of product with triplet excited carbonyl groups emitting light [16]. However, the yield of triplet excited carbonyl group formation is very low, and thus, the intensity of chemiluminescence is very small and is named UPE (ultraweak photon emission). Free Fe^3+^ ions have very low solubility at a pH of 7.4 and form precipitating Fe(OH)_3_. In our system, an excess of EGTA prevents this process, and Fe^3+-^EGTA can undergo reduction into Fe^2+^-EGTA by any reducing agent added to the Fenton system. Table 4 shows the main plausible mechanisms by which plant polyphenols may alter the UPE of the 92.6 µmol/L Fe^2+^—185.2 µmol/L EGTA—2.6 mmol/L H_2_O_2_ system. The highest tested polyphenol concentration (50 µmol/L) was 52-times lower than the concentration of H_2_O_2_ in the modified Fenton system. Therefore, the direct reaction of the tested compound with H_2_O_2_ could not significantly decrease the H_2_O_2_ pool reacting with Fe^2+^. Moreover, the incubation of each tested compound with Fe^2+^-H_2_O_2_ did not result in light emission. EGTA is an effective chelator of divalent cations, and in our experiments, Fe^2+^ ions were added to the reaction mixture after polyphenol and EGTA. Furthermore, the concentration of EGTA was 2-times higher than Fe^2+^ ions and was 3.7- to 37-times higher than that of the studied polyphenols. Taking the above into consideration, one may conclude that the decomposition of H_2_O_2_ and the chelation of Fe^2+^ ions by the studied polyphenols, as well as light emissions from polyphenols undergoing oxidation by ^•^OH radicals and H_2_O_2_ (mechanisms 1, 3 and 5 shown in Table 4), are not responsible for changes in UPE from the Fe^2+^-EGTA-H_2_O_2_ system. There was a 28-fold molar excess of H_2_O_2_ in comparison to the Fe^2+^ ions in the modified Fenton system. Therefore, the regeneration of Fe^2+^ -EGTA by the reduction of Fe^3+^ into Fe^2+^ could substantially enhance ^•^OH radical generation and, by consequence, the UPE. Conversely, the reaction of the tested compound with ^•^OH radicals decreases light emission by protecting ether bonds of EGTA from oxidative attack. Superoxide radicals (O_2_^−^) are also generated in the Fenton system [16]. They can regenerate Fe^2+^-EGTA in the following reaction:Fe^3+^-EGTA + O_2_^−^ → Fe^2+^-EGTA + O_2_

Superoxide dismutase (a highly effective O_2_^−^ radical scavenger) inhibited the UPE of the Fe^2+^-EGTA-H_2_O_2_ system [16]. Therefore, these three aforementioned processes (mechanisms 2, 4 and 6, shown in Table 4) seem to be responsible for changes in the UPE of the Fe^2+^-EGTA-H_2_O_2_ system caused by the evaluated polyphenols.

### 3.2. Phenolics That Suppressed Light Emission from the Fe^2+^-EGTA-H_2_O_2_ System within the Concentration Range of 5µmol/L to 50 µmol/L

This antioxidant activity, expressed by the suppression of light emission, can be attributed to reactions with ^•^OH and O_2_^−^ radicals that protected EGTA from peroxidative attack and that inhibited the regeneration of the Fe^2+^-EGTA complex, respectively. This is in line with the observation that mannitol and DMSO (^•^OH radical scavengers), as well as superoxide dismutase, decreased light emission from the Fe^2+^-EGTA-H_2_O_2_ system [16]. Polyphenols can react with superoxide radicals, and this includes proton-transfer (acid-base) and/or radical-transfer pathways [25].

Proton transfer mechanism:O_2_^−^ + AH → HO_2_^•^ + A^−^ (AH -polyphenol)
HO_2_^•^ + O_2_^−^→ HO_2_^−^ + O_2_
HO_2_^−^ + AH → H_2_O_2_ + A^−^

Radical transfer mechanism:O_2_^−^ + AH → HO_2_^−^ + A^•^ (A^•^-phenoxyl radical)

Phenoxyl radicals can dimerize or oligomerize to form nonradical products. Transformation into quinone or semiquinone can be another way of phenoxyl radical elimination. Both phenolic acids (benzoic acid and cinnamic acid derivatives) and polyphenols can react with superoxide, whereas the highest activity was observed for compounds containing a o-diphenol ring (e.g., flavonoids) [25]. All compounds that inhibited the UPE of Fe^2+^-EGTA-H_2_O_2_ were phenolic acids (monophenols), and one may suppose that proton transfer is a leading pathway of O_2_^−^ inactivation that was responsible for this phenomenon. However, the pH of the chemical reaction environment was stabilized with phosphate buffer (pH = 7.4), and therefore, the radical transfer mechanism of O_2_^−^ inactivation seems dominant under the conditions of our experiments. Phenolic acids with one hydroxy group in the ortho or para position to the carboxyl group were reported to be effective O_2_^−^ scavengers [26]. Moreover, compounds that possess more than one hydroxy group in their aromatic ring (such as gallic acid, caffeic acid), especially those with –OH groups in the *ortho* and *para* position to the carboxyl group, showed antioxidant properties against O_2_^−^ radicals [26,27].

Hydrogen abstraction and -OH addition to double bonds are chemical reactions involved in the inactivation of ^•^OH radicals by polyphenols [28]. ^•^OH radicals can abstract H• from -OH or -OCH_3_ substituents of a given polyphenol and can form H_2_O and phenolic radicals [28,29]. In the second type of reaction, ^•^OH radical can be added to the double bond of the phenolic ring of a polyphenol or to a double bond of an aliphatic substitute at the benzene ring. This results in the formation of phenolic hydroxy derivative radicals [28]. The presence of two adjacent (position ortho) -OH substituents (catechol structure) or one -OH and one -OCH_3_ on a benzene ring (methoxyphenol structure) facilitates reactivity with ^•^OH radicals by lowering the activation barrier for -OH addition [30]. Since the phenolic radicals did not emit light (Table 5, control experiments, sample no 4, 6 and 7), the resulting ^•^OH radicals inactivation may be responsible for the suppression of the UPE of the Fe^2+^-EGTA-H_2_O_2_ system. It should be pointed out that phenolics that revealed the highest antioxidant activity (3,4-dihydroxyphenyl-acetic acid, homovanillic acid, vanillic acid and caffeic acid) had catechol or methoxyphenol structures inside its molecule. The only exception was orthocresol, having a -CH_3_ substituent instead of -OCH_3_. Moreover, the aliphatic substitute of caffeic acid has one double bond. The remaining three phenolics (4-hydroxy phenyl acetic acid, 3-hydroxybenzoic acid and hippuric acid) had no afore-mentioned structures, and their inhibition of the UPE of Fe^2+^-EGTA-H_2_O_2_ did not exceed 50% at concentrations of 50 µmol/L and may result from the scavenging of superoxide radicals [26,27,31].

### 3.3. Phenolics That Enhanced Light Emission from the Fe^2+^-EGTA-H_2_O_2_ System within the Concentration Range of 5 µmol/L to 50 µmol/L

The regeneration of the Fe^2+^-EGTA complex by the reduction of Fe^3+^ into Fe^2+^ seems to be the most likely mechanism responsible for the polyphenol-induced enhancement of light emission from the Fe^2+^-EGTA-H_2_O_2_ system. However, the ability to reduce ferric ions is recognized as a measure of the antioxidant potential of a given compound [32], but in our experimental model, the high efficacy of Fe^2+^ regeneration leads to increased ^•^OH production being the pro-oxidant mechanism. Numerous studies on plant polyphenol-induced Fe^3+^ reduction that involved a dozen to several dozens of various compounds have been executed [8,33,34]. They showed that polyphenols can reduce Fe^3+^ to Fe^2+^, and some of them had a ferric reducing ability (FRAP), similar to that of equimolar concentrations of ascorbic acid [8]. The FRAP of polyphenols is positively correlated with the presence of catechol or a methoxyphenol ring in their molecules [8,33]. Moreover, three hydroxy groups being located in the ortho or para position to each other results in the increased ability to reduce Fe^3+^ ions [34]. The presence of -OH substituents at position 1 and other substituents at position 3 (meta position) enhances the FRAP of phenolic acids [33]. Moreover, the presence of -OH substituents at position 3 on ring C increases the FRAP of flavonoids [33]. On the other hand, phenolic acids that contain only one hydroxy group, as well as those with two hydroxy groups in the meta position, have lower activity than those containing catechol structures [34].

The reaction between the polyphenol -OH substituent (A-OH) and Fe^3+^-EGTA most likely involves the formation of complexes and hydrogen atom transfers, leading to the formation of Fe^2+^-EGTA and phenoxyl radicals (AO^•^) [34].
Fe^3+^-EGTA + A-OH → A-OH-Fe^3+^-EGTA → Fe^2+^-EGTA + AO^•^ + H^+^

The amount of Fe^2+^-EGTA produced depends on the Fe^3+^ to polyphenol ratio and on the type of compound [35,36]. Of five compounds that enhanced the UPE of the Fe^2+^-EGTA-H_2_O_2_ system (Table 2), gallic acid, ellagic acid and pelargonidin contained the aforementioned structures responsible for the high ability to reduce Fe^3+^ into Fe^2+^. On the other hand, pelargonidin-3-*O*-rutinoside, at a concentration of 5µmol/L, had lower pro-oxidant activity than pelargonidin, most likely due to the presence of a rutinosyl group at the position 3 [37]. Phloroglucinol, containing three hydroxy groups in the meta position to each other, was the second after gallic acid for the enhancing of light emission from the Fe^2+^-EGTA-H_2_O_2_ system. This contrasts with the aforementioned relationships between the chemical structures of phenolics and their ability to reduce Fe^3+^ ions. It should be noted that FRAP is measured with tripyridyltriazine under conditions of acidic milieu (pH = 3.6) [32]. In our study, chemical reactions were conducted at a pH of 7.4, and this may explain this discrepancy.

### 3.4. Polyphenols Which Altered Their Antioxidant Activity into Pro-Oxidant Activity (or Vice Versa) within the Concentration Range of 5 µmol/L to 50 µmol/L

As discussed above, the presence of catechol or methoxyphenol structures in polyphenol molecules is associated with the efficient decomposition of ^•^OH radicals and the regeneration of the Fe^2+^-EGTA complex. These processes have the opposite effect on the UPE of the Fe^2+^-EGTA-H_2_O_2_ system. The scavenging of ^•^OH radicals decreases light emission, and Fe^3+^ reduction into Fe^2+^ enhances light emission (Table 4). Three phenolics (ferulic acid, chlorogenic acid and cyanidin 3-*O*-glucoside), which altered their activity from pro-oxidant into antioxidant along with increasing concentrations, contained a catechol or methoxyphenol ring. It is possible that these phenolics can simultaneously scavenge ^•^OH radicals and reduce Fe^3+^ ions, and at higher concentrations, the first process prevails, being responsible for the antioxidant effect reflected by the inhibition of light emission from the Fe^2+^-EGTA-H_2_O_2_ system (Figure 1). This assumption can be generalized to other studied phenolics containing the aforementioned rings and can the explain non-linear relationship between compound concentrations and alterations in light emissions revealed by vanillic acid, homovanillic acid (Table 1), gallic acid, pelargonidin and pelargonidin-3-*O*-rutinoside (Table 2). However, the scavenging of O_2_^−^ radicals can be an additional factor responsible for this non-linearity in the case of gallic acid, pelargonidin and pelargonidin-3-*O*-rutinoside [27,38]. Resorcinol (1,3-dihydroxybenzene, two hydroxy groups at the benzene ring in a meta position) confirms this explanation. At concentrations of 5 and 25 µmol/L, this compound did not alter significantly, whereas at a concentration of 50 µmol/L, it enhanced the UPE of the Fe^2+^-EGTA-H_2_O_2_ system. Trolox serves as a positive control because it reveals antioxidant or pro-oxidant activities, depending on its concentrations in experiments with cell cultures [11]. Trolox efficiently reduces Fe^3+^ into Fe^2+^ ions [39] and scavenges ^•^OH radicals [40]. Moreover, the addition of Trolox into an aerated solution of Cu^2+^ ions results in the generation of ^•^OH radicals due to the reduction of Cu^2+^ into Cu^+^ [41]. Therefore, the effect of increasing concentrations of Trolox on light emission from the Fe^2+^-EGTA-H_2_O_2_ system was bidirectional and depended on the net outcome of the direct scavenging of ^•^OH radicals and on their enhanced formation due to Fe^3+^ reduction.

### 3.5. Strengths and Limitations of the Study

The concentrations of reactants used in our experimental model are close to possibly occurring in human body fluids. The plasma levels of H_2_O_2_ and iron, complexed with low molecular weight compounds, can reach 50 µmol/L and 10 µmol/L in certain diseases [42,43]. It is possible that H_2_O_2_ concentrations can be even higher in activated inflammatory cells. Moreover, 17 phenolics and Trolox were studied at 3 concentrations that can occur in the plasma of systemic circulation, portal circulation or intracellular fluid of enterocytes after eating meals rich in plant phytochemicals [15,44]. In addition, the usage of an *undeaerated* medium with pH = 7.4, a temperature of 37 °C and not being exposed to sunlight resembles in vivo conditions. These are the advantages of our study, which indicate that the observed phenomena can occur in the human body. Numerous organic buffers, including Hepes (*N*-(2-hydroxyethyl), piperazine-*N*′-(2-ethanesulfonic acid)), Tricine (*N*-[tris (hydroxymethyl)methyl] glycine) and Tris (tris(hydroxymethyl) aminomethane) can effectively scavenge ^•^OH radicals [45]. The phosphate buffer system is important in buffering intracellular fluid. These were the reasons for choosing the phosphate buffer in our experiments. However, phosphate buffer solution is able to bind Fe^2+^ and Fe^3+^ to form complexes with low solubility and reactivity, thus inhibiting ^•^OH generation by the Fenton system [46]. On the other hand, this inhibitory effect was completely abolished by the addition of EDTA, most likely by the formation of ternary complexes, chelator-Fe^2+^-phosphate [46] or chelator-Fe^3+^-phosphate. Since the EDTA chemical structure is similar, to some extent, to that of EGTA, it can be assumed that the latter has the same effect on ^•^OH formation in a medium containing phosphate buffer. ^•^OH radicals are the major ROS generated in reactions of Fe^2+^ with H_2_O_2_ only in acidic conditions, whereas high-valent oxoiron (IV) species are mainly generated when a reaction is carried out in a neutral or alkaline environment. However, in the presence of a phosphate buffer, oxoiron (IV) species are efficiently converted to ^•^OH radicals [47]. Therefore, phosphate buffers seem to facilitate ^•^OH radical generation in the Fenton reaction under conditions of physiological pH. Phosphoric acid, in contrast, can inactivate ^•^OH radicals through hydrogen atom transfer reactions [47]. These clearly show the difficulties with the selection of effective and physiologically relevant buffers to conduct the Fenton reaction in vitro. Reactions of polyphenols with O_2_^−^, ^•^OH radicals and Fe^3+^ result in the formation of various polyphenol radicals. The exposition of EGTA to ^•^OH radicals causes the cleavage of the ether bond in the backbone structure of this chelator, with further formation of peroxyl radicals that may convert into products with triplet excited carbonyl groups emitting light [16]. It cannot be excluded that polyphenols and EGTA-derived radicals can react with each other and thus decrease the UPE of Fe^2+^-EGTA-H_2_O_2_. We suggest that the presence of catechol or methoxyphenol as a key structure is responsible for polyphenol reactivity with ^•^OH radicals and Fe^3+^ ions. However, we were not able to solve two important questions: (1) What is responsible for the opposite activities (anti- and pro-oxidant) of the two phenolics (e.g., 3,4-dihydroxyphenyl-acetic acid and gallic acid) containing catechol or methoxyphenol structures?; (2) Why can the same phenolic-containing catechol structure (e.g., ferulic acid) have pro-oxidant activity at a concentration of 5 µmol/L and have antioxidant activity at a concentration of 50 µmol/L? The spatial structure of a given phenolic; its molecular mass; the presence and properties of substituents on benzene rings; and the ability to form complexes with Fe^2+^-EGTA or Fe^3+^-EGTA may be the possible determinants of these phenomena. These unresolved questions can be recognized as the limitations of our study. On the other hand, they can be an inspiration for further studies with sophisticated research protocols and equipment to precisely characterize the influence of plant phenolics on redox processes in the human body and the consequences of the dietary supplementation of these phytochemicals.

## 4. Materials and Methods

### 4.1. Reagents

All chemicals were of analytical grade. Iron (II) sulfate heptahydrate (FeSO_4_·7H_2_O), ethylene glycol-bis (β-aminoethyl ether)-N,N,N′,N′,-tetraacetic acid (EGTA) and Trolox^®^ (a water-soluble analog of vitamin E) were purchased from Sigma-Aldrich Chemical (St. Louis, MO, USA). Sterile phosphate-buffered saline (PBS, pH 7.4, without Ca^2+^ and Mg^2+^ osmolarity 300 mOsmol/L) and an H_2_O_2_ 30% solution (*w*/*w*) were obtained from Biomed (Lublin, Poland) and Chempur (Piekary Slaskie, Poland), respectively. The following plant polyphenols and their metabolites of the highest purity available were acquired from Sigma-Aldrich Chemie GmbH (Steinheim, Germany) or from Fluka, Sigma-Aldrich (Buchs, Steinheim, Germany) and were tested: 3,4-dihydroxyphenyl-acetic acid, orthocresol, homovanillic acid, vanillic acid, caffeic acid, 4-hydroxyphenyl-acetic acid, 3-hydroxybenzoic acid, hippuric acid, gallic acid, phloroglucinol, ellagic acid, ferulic acid, chlorogenic acid and resorcinol. Pelargonin chloride, pelargonidin-3-*O*-rutinoside chloride and cyanidin 3-*O*-glucoside (all chromatography standard grade) were obtained from Extrasynthese S. A. (Genay, France).

Sterile deionized pyrogen-free H_2_O (freshly prepared, resistance > 18 MW/cm, HPLC H_2_O Purification System, USF Elga, Buckinghamshire, UK) was used throughout the study.

A stock solution of EGTA (100 mmol/L) was prepared in PBS with a pH adjusted to 8.0 with 5 mol/L NaOH and was stored at room temperature in the dark for no longer than 3 months. The working solutions are as follows: 10 mmol/L EGTA (obtained by appropriate dilution of EGTA stock solution with water), 5 mmol/L FeSO_4_ in water and 28 mmol/L H_2_O_2_ (obtained by the dilution of 30% H_2_O_2_ solution with water; the concentration was confirmed by the measurement of absorbance at 240 nm, molar extinction coefficient of 43.6 mol^−1^cm^−1^ [48]), which were prepared just before use. All polyphenols and Trolox^®^ were dissolved in PBS to concentrations of 5.8, 29 and 58 µmol/L just before the experiments.

### 4.2. Emitting Light System and Measurements of Total Light Emanation

Based on our previous studies [16,17], the modified Fenton system, composed of 92.6 µmol/L Fe^2+^, 185.2 µmol/L EGTA and 2.6 mmol/L H_2_O_2_, was used for light generation. The reaction of Fe^2+^ with H_2_O_2_ causes the formation of hydroxyl radicals (^•^OH). ^•^OH radicals can oxidatively attack and cleave ether bonds in the EGTA molecule, which leads to the formation of derivatives with triplet excited carbonyl groups [16]. The decay of these excited groups results in ultra-weak photon emission (UPE) with a spectral range of 350 nm to 550 nm [49,50]. A multitube luminometer (AutoLumat Plus LB 953, Berthold, Germany), equipped with a Peltier-cooled photon counter (spectral range from 380 to 630 nm) to ensure high sensitivity and low and stable background noise signals, was used for the measurements of total light emission. Briefly, 20 µL of 10 mmol/L EGTA working solution was added to the tube (Lumi Vial Tube, 5 mL, 12 × 75 mm, Berthold Technologies, Bad Wildbad, Germany) containing 940 µL of PBS, and then 20 µL of 5 mmol/L working solution of FeSO_4_ was added. After gentle mixing, the tube was placed in the luminometer chain and was incubated for 10 min in the dark at 37 °C. Then, 100 µL of 28 mmol H_2_O_2_ solution was injected by an automatic dispenser, and the total light emission (expressed in RLU—relative light units) was measured for 120 s.

The incomplete system of Fe^2+^-H_2_O_2_ as well as Fe^2+^-EGTA without H_2_O_2_ served as controls [16,17].

### 4.3. Effect of Various Concentrations of Selected Plant Polyphenols on Light Emanation from the Fe^2+^-EGTA-H_2_O_2_ System

To assess the effect of polyphenols on the UPE of 92.6 µmol/L Fe^2+^-185.2 µmol/L EGTA-2.6 mmol/L H_2_O_2_ system, 30 µL of the working solution of the studied compounds in PBS or their appropriate dilutions was added to the luminometer tube containing EGTA and FeSO_4_ in PBS and was incubated for 10 min at 37 °C in the dark. Then, 100 µL of H_2_O_2_ solution was injected, and the total light emission was measured for 2 min. The final concentrations of a given polyphenol in the reaction mixture were 5, 25 and 50 µmol/L.

The control reagent set included: Fe^2+^-EGTA-H_2_O_2_ in PBS without polyphenol; an incomplete system of Fe^2+-^H_2_O_2_ with and without polyphenol; and polyphenol alone in PBS. The final concentration of polyphenol in the controls was 50 µmol/L. Three concentrations of the given polyphenol were tested in one series of experiments repeated at least four times. Table 5 shows the designs of these experiments. The inhibitory effect of the tested polyphenols on UPE is expressed as the percent inhibition (%I) calculated according to the formula: %I = [(A−B)/(A−C)] × 100%, where A, B and C are the total light emissions from Fe^2+^-EGTA-H_2_O_2_; Fe^2+^-EGTA-studied polyphenol -H_2_O_2_; and polyphenol alone in the medium, respectively. When UPE was augmented, the percent enhancement (%E) was calculated as follows: %E = [(B−A)/(A−C)] × 100%. We omitted measurement of the background UPE signal from the medium alone (H_2_O injected into PBS) because it was the same as those revealed by the evaluated antioxidants (including polyphenols) alone in the medium [16,17]. This shortened the experiment time and therefore reduced the risk of unexpected errors related to the extended incubation of samples in the luminometer chain.

### 4.4. Statistical Analysis

The results (total light emission, % inhibition or % enhancement of light emission) are expressed as the means (standard deviation), medians and interquartile ranges (IQR). Comparisons between the UPE of the Fe^2+^-EGTA-H_2_O_2_ system and the light emission from corresponding samples of a modified system (e.g., an incomplete system, a system with the addition of polyphenol or polyphenol in the medium alone) were analyzed with the independent-samples (unpaired) *t*-test or the Mann–Whitney U test, depending on the data distribution, which was tested with the Kolmogorov–Smirnov–Lilliefors test. The Brown–Forsythe test, used for analysis of the equality of the group variances, was used prior to the application of the unpaired *t*-test, and if variances were unequal, then the Welch’s *t*-test was used instead of the standard *t*-test. Comparisons of the % inhibition or % enhancement of UPE caused by the evaluated polyphenols were analyzed in the same way. A *p*-value < 0.05 was considered significant.

## 5. Conclusions

Plant polyphenols can act as anti- or pro-oxidants in a medium containing a modified Fenton system (Fe^2+^-EGTA-H_2_O_2_) in vitro. Distinct antioxidant activity was found for 3,4-dihydroxyphenylacetic acid, homovanillic acid, vanillic acid and caffeic acid, whereas gallic acid, phloroglucinol, pelargonidin and ellagic acid were pro-oxidant. Moreover, some of them (e.g., ferulic acid, chlorogenic acid and cyanidin 3-*O*-glucoside) can alter their activity from pro-oxidant to antioxidant (or vice versa), along with increasing concentrations from 5 µmol/L to 50 µmol/L. The presence of catechol or methoxyphenol in the backbone structures of phenolics seems to be responsible for this phenomena because they are predisposed to efficient ^•^OH radical scavenging (antioxidant activity) and to the regeneration of Fe^2+^ by the reduction of Fe^3+^ (pro-oxidant activity). The resultant effect of these processes determines the pro- or antioxidant activity of a given plant phenolic. It is assumed that this bidirectional effect of plant phenolics can occur in vivo because the concentrations of the studied phytochemicals and reagents of the Fenton system (except EGTA) are close to those of human fluids. These should be considered in the case of long-term diet supplementation with one or two polyphenols. Our results also indirectly suggest the use of a mixture of numerous plant polyphenols rather than single compounds for prophylactic dietary supplementation.

## Figures and Tables

**Figure 1 molecules-27-03453-f001:**
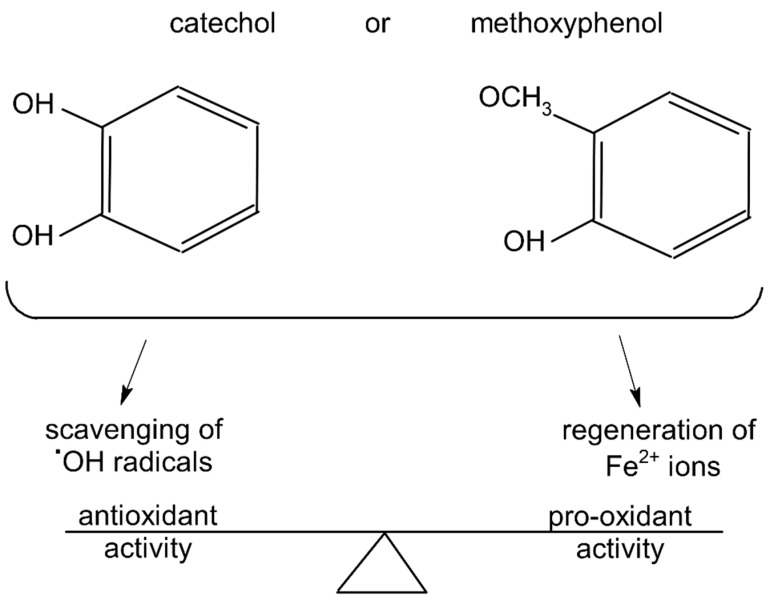
Catechol and methoxyphenol in the backbone structures of given polyphenols are responsible for scavenging of hydroxyl radicals (^•^OH) and reduction of Fe^3+^ ions into Fe^2+^ ions. Predominance of one of these processes determines anti- or pro-oxidant activity in the environment containing modified Fenton system (Fe^2+^-EGTA-H_2_O_2_).

**Table 1 molecules-27-03453-t001:** Inhibitory (antioxidant) effect of polyphenols on light emission from the 92.6 µmol/L Fe^2+^-185.2 µmol/L EGTA-2.6 mmol/L H_2_O_2_ system.

Polyphenol	Chemical Structure	% Inhibition at Concentrations of	Graph ^#^
5 µmol/L	25 µmol/L	50 µmol/L
3,4-dihydroxyphenyl-acetic acid	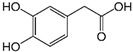	97 ± 17 (92; 18) *^†^	103 ± 21 (99; 19) *^†^	101 ± 16 (101; 14) *^†^	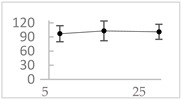
Orthocresol	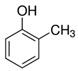	86 ± 4 (86; 3) *^†^	94 ± 1 (94; 1) *^†^	94 ± 6 (93; 6) *^†^	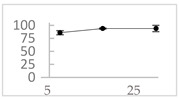
Homovanillic acid	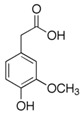	46 ± 14 (37; 15) *^‡^	85 ± 11 (86; 11) *^‡^	81 ± 13 (82; 15) *^‡^	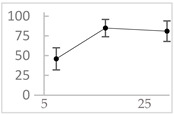
Vanillic acid	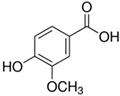	39 ± 23 (45; 24) *^‡^	70 ± 16 (73; 10) *^‡^	69 ± 3 (70; 2) *^‡^	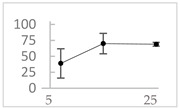
Caffeic acid	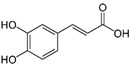	30 ± 12 (25; 11) *	66 ± 15 (63; 20) *	84 ± 20 (79;17) *	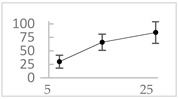
4-hydroxyphenyl acetic acid	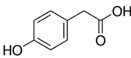	24 ± 7 (26; 5)	40 ± 7 (40; 10) *	50 ± 1 (50; 2) *	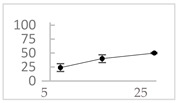
3-hydroxybenzoic acid	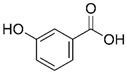	12 ± 12 (12; 15)	10 ± 13 (8; 20)	29 ± 17 (34; 10) *	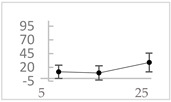
hippuric acid	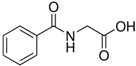	6 ± 12 (20; 19)	29 ± 14 (30; 17)	32 ± 12 (29; 14) *	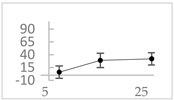

Tested compound was mixed with EGTA and Fe^2+^, and then H_2_O_2_ was automatically injected, with subsequent measurements of total light emission. Results are expressed as mean and standard deviation (median: interquartile range) of % inhibition of light emission obtained from at least four separate experiments. Orthocresol is not a typical plant phenolic, however; it is found in white cedar (*Thuja occidentalis*), birch tar and cade oils. ^#^, mean % inhibition versus concentration; *, significant inhibition, *p* < 0.05; ^†^, different from corresponding values found for vanillic acid, caffeic acid, 4-hydroxy phenyl acetic acid, 3 hydroxybenzoic acid and hippuric acid, *p* < 0.05; ^‡^, different from corresponding values found for 4-hydroxyphenyl acetic acid, 3 hydroxybenzoic acid and hippuric acid, *p* < 0.05.

**Table 2 molecules-27-03453-t002:** Enhancing (pro-oxidant) effect of polyphenols on light emission from the 92.6 µmol/L Fe^2+^-185.2 µmol/L EGTA-2.6 mmol/L H_2_O_2_ system.

Polyphenol	Chemical Structure	% Enhancement at Concentrations of			Graph ^#^
		5 µmol/L	25 µmol/L	50 µmol/L	
Gallic acid	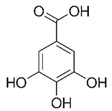	1689 ± 358 (1701; 567) *^†^	3594 ± 912 (3350; 1534) *^†^	2069 ± 484 (2128; 645) *^†^	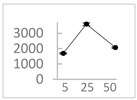
Phloroglucinol	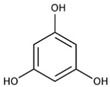	349 ± 30 (346; 46) *	1634 ± 132 (1630; 92) *^†^	2730 ± 127 (2720; 171) *^†^	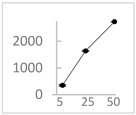
Pelargonidin	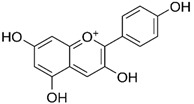	194 ± 109 (228; 92) *	187 ± 92 (227; 65) *	195 ± 98 (216; 60) *	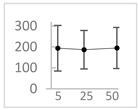
Ellagic acid	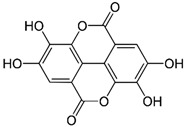	146 ± 51 (141; 85) *	398 ± 114 (366; 110) *	893 ± 180 (855; 245) *	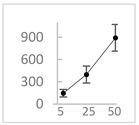
pelargonidin-3-*O*-rutinoside	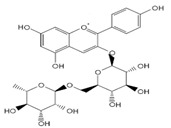	75 ± 7 (73; 9) *	264 ± 50 (265; 86) *	258 ± 28 (260; 38) *	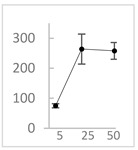

Tested compound was mixed with EGTA and Fe^2+^, and then H_2_O_2_ was automatically injected, with subsequent measurements of total light emission. Results are expressed as mean and standard deviation (median: interquartile range) of % enhancement of light emission obtained from at least four separate experiments. ^#^, mean % enhancement versus concentration; *, significant enhancement, *p* < 0.05; ^†^, different from corresponding values found for pelargonidin, ellagic acid and pelargonidin-3-*O*-rutinoside, *p* < 0.05.

**Table 3 molecules-27-03453-t003:** Polyphenols which altered their antioxidant activity into pro-oxidant activity (or vice versa) within the concentration range of 5 µmol/L to 50 µmol/L, as evaluated with the light emitting system: 92.6 µmol/L Fe^2+^-185.2 µmol/L EGTA-2.6 mmol/L H_2_O_2_.

Polyphenol	Chemical Structure	% Inhibition (↓) or Enhancement (↑) at Concentrations of			Graph ^#^
		5 µmol/L	25 µmol/L	50 µmol/L	
Ferulic acid	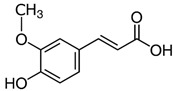	↑63 ± 22 (65; 35) *	↑14 ± 13 (13; 18)	↓28 ± 10 (28; 14) *	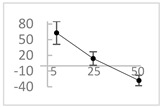
Chlorogenic acid	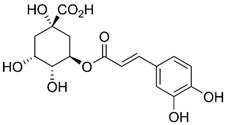	↑5 ± 22 (4; 22)	↓78 ± 5 (78; 8) *	↓94 ± 3 (94; 4) *	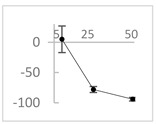
cyanidin3-*O*-glucoside	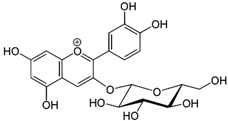	↑445 ± 65 (322; 97) *^†^	↑80 ± 35 (64; 47) *^‡^	↓24 ± 14 (23; 21)	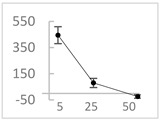
Trolox	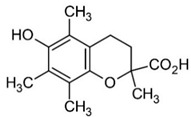	↑479 ± 51 (505; 45) *^†^	↓104 ± 7 (100; 6) *	↓105 ± 10 (99; 6) *	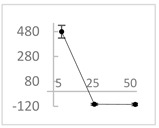
Resorcinol	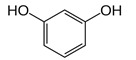	↓ 22 ± 17 (29; 24)	↑10 ± 19 (6; 16)	↑53 ± 23 (47; 19) *	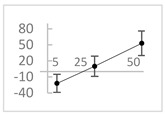

The tested compound was mixed with EGTA and Fe^2+^, and then H_2_O_2_ was automatically injected, with subsequent measurements of total light emission. Results are expressed as mean and standard deviation (median: interquartile range) of % enhancement (↑) or % inhibition (↓) of light emission obtained from at least four separate experiments. ^#^, mean % enhancement (positive values) or % inhibition (negative values) versus concentration. Resorcinol is not a typical plant phenolic, however; it is found in broad bean (*Vicia faba*) and in argan oil and is extracted from fruit kernels of argan trees (*Argania spinosa*). Trolox is a water-soluble analog of vitamin E. *, significant enhancement or inhibition, *p* < 0.05; ^†^, different from corresponding values found for ferulic acid and chlorogenic acid; *p* < 0.05; ^‡^, different from corresponding values found for ferulic acid and resorcinol. Both % inhibition and % enhancement are referred to with the base value noted for Fe^2+^-EGTA-H_2_O_2_ alone (without any studied compound addition).

**Table 4 molecules-27-03453-t004:** Plausible mechanisms by which polyphenols may affect light emission from the 92.6 µmol/L Fe^2+^-185.2 µmol/L EGTA-2.6 mmol/L H_2_O_2_ system.

Proposed Mechanism of Action	Effect on UPE
1. Reaction with H_2_O_2_	Suppression
2. Reaction with ^•^OH radicals	Suppression
3. Chelation of Fe^2+^ ions to form less reactive complexes	Suppression
4. Regeneration of Fe^2+^ ions by the reduction of Fe^3+^ into Fe^2+^	Enhancement
5. Reaction with H_2_O_2_ or ^•^OH radicals to form products that emit light	Enhancement
6. Reaction with O_2_^−^ radicals	Suppression

UPE, ultra-weak photon emission.

**Table 5 molecules-27-03453-t005:** Design of experiments on the effects of selected polyphenols at concentrations of 5, 25 and 50 µmol/L on light emission from the Fe^2+^-EGTA-H_2_O_2_ system.

No.	Sample	Volumes of Working Solutions Added to Luminometer Tube (µL)					
		A	B	C	D	E	F
		PBS	Polyphenol *	EGTA	FeSO_4_	H_2_O_2_	H_2_O
1	Complete system	940	-	20	20	100	-
2	Complete system + polyphenol	-	940	20	20	100	-
3	Incomplete system	960	-	-	20	100	-
4	Incomplete system + polyphenol	20	940	-	20	100	-
Additional controls							
5	Fe^2+^-EGTA without H_2_O_2_	940	-	20	20	-	100
6	Fe^2+^-EGTA without H_2_O_2_ + Polyphenol	-	940	20	20	-	100
7	Polyphenol alone	40	940	-	-	-	100

Working solutions were mixed in alphabetical order: A, sterile phosphate buffered saline (PBS, pH = 7.4) without divalent cations; B, polyphenol solution in PBS (concentrations of 5.8, 29 or 58 µmol/L); C, 10 mmol/L aqueous solution of EGTA; D, 5 mmol/L agueous solution of FeSO_4_. Then, after gentle mixing, the tube was placed into a luminometer chain and incubated for 10 min at 37 °C. Then, 28 mmol/L H_2_O_2_ (E) or H_2_O (F) was automaticly injected with a dispenser, and the total light emission was measured for 2 min. *, in some experiments, Trolox solution was was added instead of polyphenols.

## Data Availability

The data presented in this study are available on request from the corresponding author.

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
