# Peer review of "Concentration Dependence of Anti- and Pro-Oxidant Activity of Polyphenols as Evaluated with a Light-Emitting Fe2+-Egta-H2O2 System"

_molecules, 2022, doi:10.3390/molecules27113453_

Round 1
Reviewer 1 Report
Manuscript title is too long. Authors should consider shortening the title. The abstract meets all scientific and technical criteria. The potential application of the obtained results and the importance of the research should be emphasized in more detail. The chemical structure of pelargonidine-3-Orutinoside in Table 2 needs to be improved technically, it is currently of poor quality. The image size of Phloroglucinol should be reduced, equalized with the others in the table. Do the same for Table 3. The idea of the research, selected methods, results and their discussion meet all scientific and technical criteria. The conclusion should be written in more detail based on all the results obtained. The Latin names of the plants in the references should be italic.
Author Response
Response to comments of the Reviewer 1
1 .Manuscript title is too long. Authors should consider shortening the title.
Response (Re) : the shortened title is as follows: ‘Concentration dependence of anti- and pro-oxidant activity of polyphenols as evaluated with light emitting Fe2+-EGTA-H2O2 system”.
- 2. The chemical structure of pelargonidine-3-Orutinoside in Table 2 needs to be improved technically, it is currently of poor quality. The image size of Phloroglucinol should be reduced, equalized with the others in the table. Do the same for Table 3.
Re: These are corrected according to the Reviewer 1 suggestion.
- 3. The conclusion should be written in more detail based on all the results obtained
Re: We modified the Conclusions section and now is as follows:
“Plant polyphenols can act as anti- or pro-oxidants in medium containing modified Fenton’s system (Fe2+-EGTA-H2O2 ) in vitro. Distinct antioxidant activity was found for 3,4-dihydroxyphenylacetic acid, homovanillic acid, vanillic acid and caffeic acid while gallic acid, phloroglucinol, pelargonidin and ellagic acid were pro-oxidant. Moreover, some of them (e.g. ferulic acid, chlorogenic acid and cyanidin 3-O-glucoside) can alter their activity from pro-oxidant to antioxidant (or vice versa) along with the increasing concentrations from 5 µmol/L to 50 µmol/L. Presence of catechol or methoxyphenol in the backbone structure of phenolics seems to be responsible for this phenomena because they predispose to efficient •OH radicals scavenging (antioxidant activity) and regeneration of Fe2+ by reduction of Fe3+ (pro-oxidant activity). The resultant effect of these processes determines pro- or antioxidant activity of given plant phenolic. It is supposed that this bidirectional effect of plant phenolics can occur in vivo because of concentrations of studied phytochemicals and reagents of Fenton’s system (except of EGTA) are close to those of human fluids. These should be considered in the case of long-term diet supplementation with one or two polyphenols. Our results also indirectly suggest use of mixture of numerous plant polyphenols rather than single compounds for prophylactic dietary supplementation.”
- 4. The Latin names of the plants in the references should be italic.
Re: This is corrected throughout the entire manuscript (including the references)
All changes in the text are marked in red.
Reviewer 2 Report
The paper "Concentration Dependence of anti- and pro-oxidant Activity of Selected Polyphenols as Evaluated with Hydroxyl Radical-induced Light Emission from Fe2+-EGTA-H2O2 System" is an excellent study of the influence of one and a half dozen of nature occurred polyphenols on the hydroxyl radical generation in the Fenton system. The thorough experiments revealed that some of the studied compounds are able to enhance the yield of ROS, while other ones decrease the generation. Few compounds were found to change their action depending on the concentration of polyphenol.
Authors made plausible and sound explanation to all the phenomena they observed. The accurate statistical analysis made to the data deserves a special praise; it is rare to see such efforts nowadays.
Some minor points, however, are yet to be addressed, IMHO:
- Authors give an asterisk in Tables 1-3 to those values, which are the significant enhancement or inhibition. Why the values in Table 1 for homovanilinic acid went unnoticed? All of them exceed the last value for the hippuric acid, which was highlighted.
- I recommend adding the error bars to all the graphs.
- It is not completely clear in Table 3, to what level are referred to the values of the inhibition (in the cases when inhibition follows the enhancement). To the base level or to the new enhanced level. It does not matter for trolox because 100% inhibition is a complete inhibition no matter what is the starting point, but it matters for other compounds.
- Both Fe2+ and Fe3+ ions can form the complexes with phosphate from buffer (and phosphate, if I understood in correctly, presents in a high concentration), and the stability of these complexes may differ by 5 log units depending on the oxidation state. The complexes of Fe3+ with EGTA should also be more stable by order of 8-9, I believe, than those formed by Fe2+. The chemical equilibria between different iron species are quite difficult in the system under study. It is an additional probable reason for your Discussion section 3.5.
- In the Table 1, in hippuric acid formula, N looks like cyrillic И :)
Author Response
Response to comments of the Reviewer 2
- 1. Authors give an asterisk in Tables 1-3 to those values, which are the significant enhancement or inhibition. Why the values in Table 1 for homovanilinic acid went unnoticed? All of them exceed the last value for the hippuric acid, which was highlighted.
Response (Re): It was just an oversight. This is corrected in the revised manuscript
- 2. I recommend adding the error bars to all the graphs.
Re: These are corrected according to the Reviewer 2 suggestion.
- 3. It is not completely clear in Table 3, to what level are referred to the values of the inhibition (in the cases when inhibition follows the enhancement). To the base level or to the new enhanced level. It does not matter for trolox because 100% inhibition is a complete inhibition no matter what is the starting point, but it matters for other compounds.
Re: This was always referred to the base level observed without studied compound addition. We clarified this issue in the legend of the Table 3 by adding the following statement: “ Both, % inhibition or % enhancement are referred to the base value noted for Fe2+ -EGTA- H2O2 alone (without any studied compound addition).”
- 4. Both Fe2+and Fe3+ions can form the complexes with phosphate from buffer (and phosphate, if I understood in correctly, presents in a high concentration), and the stability of these complexes may differ by 5 log units depending on the oxidation state. The complexes of Fe3+ with EGTA should also be more stable by order of 8-9, I believe, than those formed by Fe2+. The chemical equilibria between different iron species are quite difficult in the system under study. It is an additional probable reason for your Discussion section 3.5.
Re: Discussion section 3.5 (Strengths and limitations of the study) - We added the following part: “Numerous organic buffers including Hepes (N-(2-hydroxyethyl) piperazine-N´-(2-ethanesulfonic acid)), Tricine (N-[tris (hydroxymethyl)methyl] glycine) and Tris (tris(hydroxymethyl) aminomethane) can effectively scavenge •OH radicals [45]. Phosphate buffer system is important in buffering intracellular fluid. Those were the reasons for choosing the phosphate buffer in our experiments. However, phosphate buffer solution is able to bind Fe2+ and Fe3+ to form complexes with low solubility and reactivity and thus inhibit •OH generation by Fenton system [46]. On the other hand, this inhibitory effect was completely abolished by addition of EDTA probably by formation of ternary complexes; chelator-Fe2+-phosphate [46] or chelator-Fe3+-phosphate. Since EDTA chemical structure is similar to some extent to that of EGTA it can be assumed that the latter would have the same effect on •OH formation in medium containing phosphate buffer. •OH radicals are the major ROS generated in reaction of Fe2+ with H2O2 only in acidic conditions whereas high-valent oxoiron (IV) species are mainly generated when reaction is carried out in neutral or alkaline environment. However, in the presence of phosphate buffer oxoiron (IV) species are efficiently converted to •OH radicals [47]. Therefore phosphate buffers seem to facilitate •OH radicals generation in the Fenton reaction under conditions of physiological pH. Phosphoric acid, in contrast, can inactivate •OH radicals through hydrogen atom transfer reactions [47]. These clearly shows the difficulties with selection of the effective and physiologically relevant buffer to conduct the Fenton reaction in vitro.” The number of cited references increased by 3 up to 50.
- In the Table 1, in hippuric acid formula, N looks like cyrillic И :)
Re: This was corrected.
All changes in the text are marked in red.